# Effects of realistic e-learning cases on students' learning motivation during COVID-19

Ann-Kathrin Rahm[1,2]*, Maximilian Töllner[1], Max Ole Hubert[1], Katrin Klein[3], Cyrill Wehling[4], Tim Sauer[5], Hannah Mai Hennemann[5], Selina Hein[2], Zoltan Kender[6], Janine Günther[5], Petra Wagenlechner[7], Till Johannes Bugaj[7], Sophia Boldt[7], Christoph Nikendei[7], Jobst-Hendrik Schultz[1,7]

1 Heidelberg University Medical School, Heidelberg, Germany, 2 Department of Cardiology, Heidelberg University Hospital, Heidelberg, Germany, 3 Department of Nephrology, Heidelberg University Hospital, Heidelberg, Germany, 4 Department of Gastroenterology, Infectious Diseases and Intoxication, Heidelberg, Germany, 5 Department of Hematology, Heidelberg University Hospital, Heidelberg, Germany, 6 Department of Endocrinology, Diabetology and Clinical Chemistry, Heidelberg University Hospital, Heidelberg, Germany, 7 Department of General Internal Medicine and Psychosomatics, Heidelberg University Hospital, Heidelberg, Germany

☯ These authors contributed equally to this work.
* ann-kathrin.rahm@med.uni-heidelberg.de

**Data Availability Statement:** The original German free-text students' answers to the questionnaires are not directly presented to protect the identities of participants. However, all relevant data required

## Abstract

### Background

Keeping up motivation to learn when socially isolated during a pandemic can be challenging. In medical schools, the COVID-19 pandemic required a complete switch to e-learning without any direct patient contact despite early reports showing that medical students preferred face-to-face teaching in clinical setting. We designed close to real-life patient e-learning modules to transmit competency-based learning contents to medical students and evaluated their responses about their experience.

### Methods

Weekly e-learning cases covering a 10-week leading symptom-based curriculum were designed by a team of medical students and physicians. The internal medicine curriculum (HeiCuMed) at the Heidelberg University Medical School is a mandatory part of clinical medical education in the 6th or 7th semester. Case-design was based on routine patient encounters and covered different clinical settings: preclinical emergency medicine, in-patient and out-patient care and follow-up. Individual cases were evaluated online immediately after finishing the respective case. The whole module was assessed at the end of the semester. Free-text answers were analyzed with MaxQDa following Mayring's principles of qualitative content analyses.

### Results

N = 198 students (57.6% female, 42.4% male) participated and 1252 individual case evaluations (between 49.5% and 82.5% per case) and 51 end-of-term evaluations (25.8% of

to replicate the study's findings are within the paper and its Supporting Information files.

**Funding:** AKR is currently supported by the clinician-scientist-program of the German Internal Medicine Society (DGIM).

**Competing interests:** Regarding competing interests by performing this work for my Master of Medical Education degree I am happy to confirm that this does not alter our adherence to PLOS ONE policies on sharing data and materials. On behalf of all authors I can declare that no other competing interests are present.

**Abbreviations:** HeiCuMed, Heidelberg Curriculum Medicinale.

students) were collected. Students highly appreciated the offer to apply their clinical knowledge in presented patient cases. Aspects of clinical context, interactivity, game-like interface and embedded learning opportunities of the cases motivated students to engage with the asynchronously presented learning materials and work through the cases.

## Conclusions

Solving and interpreting e-learning cases close to real-life settings promoted students' motivation during the COVID-19 pandemic and may partially have compensated for missing bedside teaching opportunities.

## Background

For medical students, learning in the clinical environment by interacting with patients and physicians is essential to learn theoretical concepts and clinical workflows [1, 2]. While individual support of teachers to students for both theoretical and practical aspects of clinical activities is fundamental for successful learning [3], early patient contact in medical school stimulates students' motivation and learning [4].

Teaching during a global pandemic may therefore be challenging because clinical settings such as bedside teaching can no longer be conducted over students' and patients'safety concerns [5–8]. Early reports during the COVID-19 pandemic showed that students did not prefer e-learning over face-to-face teaching [9] and most students preferred returning to the clinical setting [10]. Online learning can show advantage by using student-centered approaches to facilitate educational access. Despite all the advantages, online learning is not the ultimate solution to all academic issues as in medical and other health professional courses, the main drawback of e-learning is in most occasions the impossibility to practice live [11].

For effectiveness of online learning principles of digital learning, goals, and student's preferences should be taken into consideration [12]. Various studies have assessed preparedness for purely e-Learning in different countries. In an analysis in Lybia only twelve percent of students agreed that e-learning could be used for clinical aspects and students were concerned about how e-learning could be applied to provide clinical experience [13]. Regarding social distancing and sole online learning effects on students'motivation are also important under these circumstances [14, 15]. Motivation can be defined as a psychological state or internal process that provides a meaning and energy to get students to act, develop and perform [16]. Motivation is positively associated with learning, performance and well-being, and diminished motivation may have deleterious effects [17]. Therefore, understanding and stabilizing motivational factors for students in (e-)learning environments becomes even more pressing in the current situation.

At Heidelberg University Medical School, the internal medicine curriculum [18–20], which is based on a 10-week leading symptom-based structure, was switched from in-presence teaching to a complete e-learning-based model. To include assessment of learning as a model for assessment-driven learning [21], ten e-learning cases for each leading symptom-based week were designed as a learning control and trigger and a potential replacement for bedside teaching. Evaluation of each case and all cases in summary at the end was performed online. Students assessed the effect of the cases on their learning time, the case's performance as learning control and effects of the cases on their learning by online questionnaires (see S1 Graphical abstract).

This study aims to enrich our understanding of how students perceive realistic multimodal, game-like e-learning cases within a complete e-learning-based curriculum. We also determined how presented aspects of the clinical environment and routine affected student's motivation and self-rated time for learning.

## Methods

### Study design

This study was designed as a single-center descriptive analysis of students' voluntary evaluations of the presented e-learning cases. Supplementary details can be found in the only supplement (S1 Text).

### Sample

All students in the internal medicine curriculum at Heidelberg University Medical School from April to July 2020 participated in the e-learning cases. Participation in online based evaluations (SurveyMonkey) of individual cases and end of term evaluation was voluntary. Therefore, sample size for this qualitative and descriptive analysis arose from students available during summer semester 2020. As performing the cases was mandatory all students were invited to take part in the facultative evaluations directly after the cases. A facultative evaluation of all cases in retrospect was performed at the end of semester voluntarily.

### Ethical considerations

An ethics approval was obtained from Heidelberg University ethics committee (S-712/2020). Individual consent was not requested since participation was voluntary and anonymous.

### Setting

Due to the COVID-19 pandemic, the internal medicine curriculum at Heidelberg University Medical School (HeiCuMed) was modified into e-learning-based teaching for summer semester 2020. The semester´s core structure of leading symptom-based lectures (S1 Table) was adopted from its previous form. Lectures and seminars were recorded and asynchronously made available for students. Further information on the curriculum structure and available materials is available as (S1 Text, S1 Table).

### Case design

E-learning cases were designed with Articulate (articulate.com). For each symptom-based learning week, a clinical case was presented to the students with various quiz and interaction modules. Each case consisted of gamification elements that required the student to apply practical and diagnostic skills and tools to proceed ahead with the patient. Additional case-specific comments of senior physicians were included as learning hints. Performing one case every week in the 10-week course within a timeframe from Friday 12am (S1 Table) till next Monday was mandatory for students to be able to register for the final exam.

To achieve a learner-centered design, cases were developed by a team consisting of medical students having already completed the internal medicine module and physicians from different disciplines providing clinical expertise and clinical examples. The underlying philosophy of all cases was to create a setting which was as close as possible to the real-life situation of patient care. Students were therefore confronted with realistic examination results, original diagnostic outcomes and direct patient interactions in order to challenge them in a more appealing way. The aim was to especially train decision-making skills, communication

behavior and diagnostic thinking. Detailed information on case design and links to work through two translated exemplary cases (S1 Share Link and S2 Share Link) and two commented video files (S1 and S2 Videos) can be found in the online supplements.

### Evaluation—Questions after each case—global rating

After completing the mandatory patient cases, students were asked to voluntarily evaluate the e-learning cases with a 10-item questionnaire. Evaluation was facultative and performed anonymously via SurveyMonkey. Six items were questions based on a 5-point Likert scale (1 = strongly disagree to 5 = strongly agree) and related to structure and style, ease of navigation, senior physicians' comments for deepening knowledge and tips and tricks, performance of different quiz styles and the aspect of enjoyment while performing the cases.

### Evaluation—Questions after each case–free text answers

Four items of the 10-item questionnaire after each individual case were free-text questions asked in a style of "what did you like about the case?", "how can we improve it further?". The term "motivation" was deliberately not used within the questions in order to not influence students' free-text answers. All German and translated to English questionnaires for individual cases can be found in the supplement section (S2 and S3 Texts).

### End of term evaluation

A final evaluation of the 10 cases was performed at the end of the semester after the final internal medicine examination. The final evaluation included global rating questions and free-text answers regarding motivation, time spent for learning within the cases and learning impulses triggered by cases. Three reminder emails were sent to students for participating in this facultative survey. The German questionnaire and its translated to English version are available in the supplement section (S2 and S3 Texts).

### Data analysis and statistics

Data analysis was performed with OriginPro2020 and MaxQDA2020 (VERBI GmbH). As free-text answers were already in written form, no transcription was needed. Texts obtained from SurveyMonkey results were checked for information that could identify an individual student. As no individuals could be identified, the interviews were subjected to a qualitative content analysis following Mayring's principles of inductive category development [22]. First, we undertook an open coding of all free-text answer questions to identify possible recurring topics. Next, individual sentences or passages were identified as one code, representing the most elementary unit of the resulting protocol [23]. Exemplary students' quotes were translated into English and embedded within the text. Using the software MaxQDA (version 2020, VERBI Software—Consult—Social Research GmbH, Berlin), we summarized individual codes as relevant topics for each case. Then, analyzers compared recurring topics from the individual cases results and assigned them to higher-level categories. The respective codes and topics were discussed to reach consensus (investigator triangulation). Finally, we subsumed the topics into a total of $n = 6$ relevant categories. We applied the categories to all transcripts using the software MaxQDA.

# Results

## Participation in facultative evaluations

All students ($n$ = 198) during the 10-week internal medicine curriculum worked through the mandatory 10 cases. Participation in facultative evaluation dropped from 164 (82.8%) in the beginning of the semester (Case 1 –chest pain) to 98 (49.5%) at the end of semester (Case 10 – musculoskeletal pain). Length of free-text answers declined from Case 1 to 10 and switched to the undirected questions from aspects of learning and motivation to mainly content-based comments.

## Factors promoting engagement

Students evaluated the cases as appealing in design (Table 1). Navigation within the first 3 cases worked which led to adaption of the questions' content during the 10-week curriculum. In most cases, the included quiz modes performed with minor issues depending on the used computer system (Table 1). The embedded senior physicians' comments for deepening knowledge and understanding were highly rated by the students. Seven out of 10 cases were evaluated as a good closure of the symptom-based weeks. In cases 3, 8 and 9 discrepancies arose from different learning foci within cases and symptom-based lectures. The demand of cases was ranked as appropriate by the students. In all 6 cases evaluated for the aspect of "fun", most students evaluated fun with average 4.1 to 4.6 on the global rating scale.

## Factors promoting motivation

Free-text coding in individual cases in open-ended questions highlighted five main topics: Interactivity, media and design, repetition and deepening of knowledge, practical aspects and fun to stimulate motivation.

**Interactivity** was highlighted by students as a factor that helped to engage with the cases (Student 49 (S49), Case 1 (C1) "everything is interactive, so I can better recall information") and stay motivated (S104, C2 "super interactive, motivates me to think on myself"). Interactive tasks were mandatory during the cases and promoted critical reflection (S8, C8 "the interactive task to identify the aortic and mitral valve [within the video] was thrilling and demanding"). Students appreciated the interactive design and wished for more (S13, C1 "The interactive design is super and I hope that much more time and resources will be invested in this concept. Excellent start").

**Media and design** were described by most students in free-text answers and cases were rated highly appreciated for realistic appearances (S21, C5 "different modi, text messages, one case over a longer period, realistic setting"). Different modi and media appeared appealing (S19, C1 "Design is super appealing and I like how many different media were put in"; S75, C2 "Graphical presentation is excellent, even handwriting notes how doctors would write them").

Regarding **learning** aspects, students found the cases to be a good revision exercise and learning control for the week´s topics that had to be learnt. (S36, C1 "A repetition of relevant facts of the week so I know where to lay my focus on"; S101, C1 "A possibility to actively use gained knowledge. The case was a good way to see what I should repeat and what's already learned" S20, C2 "Especially after learning and preparing the contents of the lectures alone at home the cases are a good learning success control alone or in groups of two/three"). Students were motivated to think critically and to put the gained knowledge into clinical context of diseases (S48, C3 "Thinking outside the box is asked"; S7, C7 "practical use of knowledge is needed to put e.g. laboratory parameters in context to diseases"). Students also noted that the cases summarized the learning context of the week's lectures. (S12, C3 "in the last weeks

**Table 1. Case-to-case evaluation by students for structure, navigation, quiz functions, performance as learning control, helpfulness of senior physician's comments, aspect of fun and demand.**

**Case Design**

| Case | Students answering | Likert Scale | | | | | | | | | | Weighted average |
|---|---|---|---|---|---|---|---|---|---|---|---|---|
| | n | 1 | | 2 | | 3 | | 4 | | 5 | | |
| | | n | % | n | % | n | % | n | % | n | % | |
| Case 1 | 164 | 0 | 0.0 | 0 | 0.0 | 0 | 0.0 | 21 | 12.8 | 144 | 87.8 | 4.9 |
| Case 2 | 154 | 0 | 0.0 | 0 | 0.0 | 2 | 1.3 | 51 | 33.1 | 101 | 65.6 | 4.7 |
| Case 3 | 139 | 3 | 2.2 | 7 | 5.0 | 41 | 29.5 | 45 | 32.4 | 43 | 30.9 | 3.9 |
| Case 4 | 141 | 0 | 0.0 | 1 | 0.7 | 12 | 8.5 | 40 | 28.4 | 88 | 62.4 | 4.5 |
| Case 5 | 106 | 0 | 0.0 | 3 | 2.8 | 21 | 19.8 | 44 | 41.5 | 38 | 35.8 | 4.1 |
| Case 6 | 121 | 0 | 0.0 | 1 | 0.8 | 10 | 8.3 | 39 | 32.2 | 71 | 58.7 | 4.5 |
| Case 7 | 117 | 0 | 0.0 | 2 | 1.7 | 6 | 5.1 | 31 | 26.5 | 78 | 66.7 | 4.6 |
| Case 8 | 103 | 1 | 1.0 | 4 | 3.9 | 17 | 16.5 | 33 | 32.0 | 48 | 46.6 | 4.2 |
| Case 9 | 99 | 0 | 0.0 | 5 | 5.1 | 15 | 15.2 | 39 | 39.4 | 40 | 40.4 | 4.2 |
| Case 10 | 98 | 0 | 0.0 | 2 | 2.0 | 12 | 12.2 | 31 | 31.6 | 53 | 54.1 | 4.4 |

**Navigation**

| Case | Students answering | Likert Scale | | | | | | | | | | Weighted average |
|---|---|---|---|---|---|---|---|---|---|---|---|---|
| | n | 1 | | 2 | | 3 | | 4 | | 5 | | |
| | | n | % | n | % | n | % | n | % | n | % | |
| Case 1 | 166 | 0 | 0.0 | 1 | 0.6 | 5 | 3.0 | 25 | 15.1 | 135 | 81.3 | 4.8 |
| Case 2 | 153 | 0 | 0.0 | 0 | 0.0 | 2 | 1.3 | 20 | 13.1 | 131 | 85.6 | 4.8 |
| Case 3 | 139 | 0 | 0.0 | 0 | 0.0 | 3 | 2.2 | 16 | 11.5 | 120 | 86.3 | 4.8 |

**Performance of quiz modi**

| Case | Students answering | Likert Scale | | | | | | | | | | Weighted average |
|---|---|---|---|---|---|---|---|---|---|---|---|---|
| | n | 1 | | 2 | | 3 | | 4 | | 5 | | |
| | | n | % | n | % | n | % | n | % | n | % | |
| Case 1 | 166 | 1 | 0.6 | 7 | 4.2 | 29 | 17.5 | 37 | 22.3 | 92 | 55.4 | 4.3 |
| Case 2 | 155 | 0 | 0.0 | 2 | 1.3 | 7 | 4.5 | 22 | 14.2 | 124 | 80.0 | 4.7 |
| Case 3 | 139 | 0 | 0.0 | 1 | 0.7 | 3 | 2.2 | 13 | 9.4 | 122 | 87.8 | 4.8 |
| Case 4 | 141 | 0 | 0.0 | 9 | 6.4 | 22 | 15.6 | 44 | 31.2 | 66 | 46.8 | 4.2 |
| Case 5 | 106 | 0 | 0.0 | 2 | 1.9 | 12 | 11.3 | 18 | 17.0 | 74 | 69.8 | 4.6 |
| Case 6 | 122 | 3 | 2.5 | 3 | 2.5 | 14 | 11.5 | 29 | 23.8 | 73 | 59.8 | 4.4 |
| Case 7 | 118 | 2 | 1.7 | 1 | 0.8 | 3 | 2.5 | 13 | 11.0 | 99 | 83.9 | 4.8 |
| Case 8 | 103 | 2 | 1.9 | 2 | 1.9 | 14 | 13.6 | 34 | 33.0 | 51 | 49.5 | 4.3 |
| Case 9 | 99 | 0 | 0.0 | 1 | 1.0 | 12 | 12.1 | 25 | 25.3 | 61 | 61.6 | 4.5 |
| Case 10 | 98 | 0 | 0.0 | 4 | 4.1 | 4 | 4.1 | 20 | 20.4 | 70 | 71.4 | 4.6 |

**Performance as learning control**

| Case | Students answering | Likert Scale | | | | | | | | | | Weighted average |
|---|---|---|---|---|---|---|---|---|---|---|---|---|
| | n | 1 | | 2 | | 3 | | 4 | | 5 | | |
| | | n | % | n | % | n | % | n | % | n | % | |
| Case 1 | 166 | 1 | 0.6 | 3 | 1.8 | 8 | 4.8 | 41 | 24.7 | 113 | 68.1 | 4.6 |
| Case 2 | 155 | 0 | 0.0 | 1 | 0.6 | 5 | 3.2 | 28 | 18.1 | 121 | 78.1 | 4.7 |
| Case 3 | 139 | 8 | 5.8 | 14 | 10.1 | 46 | 33.1 | 33 | 23.7 | 38 | 27.3 | 3.6 |
| Case 4 | 141 | 0 | 0.0 | 3 | 2.1 | 11 | 7.8 | 45 | 31.9 | 82 | 58.2 | 4.5 |
| Case 5 | 106 | 1 | 0.9 | 1 | 0.9 | 24 | 22.6 | 25 | 23.6 | 55 | 51.9 | 4.3 |

*(Continued)*

**Table 1.** (Continued)

| Case | n | n | % | n | % | n | % | n | % | n | % | WA |
|---|---|---|---|---|---|---|---|---|---|---|---|---|
| Case 6 | 122 | 0 | 0.0 | 4 | 3.3 | 7 | 5.7 | 29 | 23.8 | 82 | 67.2 | 4.6 |
| Case 7 | 118 | 0 | 0.0 | 0 | 0.0 | 10 | 8.5 | 22 | 18.6 | 86 | 72.9 | 4.6 |
| Case 8 | 103 | 2 | 1.9 | 10 | 9.7 | 18 | 17.5 | 33 | 32.0 | 40 | 38.8 | 4.0 |
| Case 9 | 98 | 3 | 3.1 | 10 | 10.2 | 20 | 20.4 | 29 | 29.6 | 36 | 36.7 | 3.9 |

**Helpfulness of senior physician's comments**

| Case | Students answering n | 1 n | 1 % | 2 n | 2 % | 3 n | 3 % | 4 n | 4 % | 5 n | 5 % | Weighted average |
|---|---|---|---|---|---|---|---|---|---|---|---|---|
| Case 1 | 162 | 1 | 0.6 | 1 | 0.6 | 10 | 6.2 | 43 | 26.5 | 107 | 66.0 | 4.6 |
| Case 2 | 154 | 0 | 0.0 | 1 | 0.6 | 2 | 1.3 | 27 | 17.5 | 124 | 80.5 | 4.8 |
| Case 3 | 139 | 1 | 0.7 | 3 | 2.2 | 14 | 10.1 | 43 | 30.9 | 78 | 56.1 | 4.4 |
| Case 4 | 141 | 0 | 0.0 | 6 | 4.3 | 12 | 8.5 | 38 | 27.0 | 85 | 60.3 | 4.4 |
| Case 5 | 106 | 0 | 0.0 | 1 | 0.9 | 7 | 6.6 | 16 | 15.1 | 82 | 77.4 | 4.7 |
| Case 6 | 122 | 0 | 0.0 | 2 | 1.6 | 6 | 4.9 | 24 | 19.7 | 90 | 73.8 | 4.7 |
| Case 7 | 118 | 0 | 0.0 | 2 | 1.7 | 5 | 4.2 | 25 | 21.2 | 86 | 72.9 | 4.7 |
| Case 8 | 102 | 2 | 2.0 | 1 | 1.0 | 7 | 6.9 | 33 | 32.4 | 59 | 57.8 | 4.4 |
| Case 9 | 99 | 1 | 1.0 | 0 | 0.0 | 7 | 7.1 | 25 | 25.3 | 66 | 66.7 | 4.6 |
| Case 10 | 98 | 0 | 0.0 | 1 | 1.0 | 8 | 8.2 | 22 | 22.4 | 67 | 68.4 | 4.6 |

**Fun**

| Case | Students answering n | 1 n | 1 % | 2 n | 2 % | 3 n | 3 % | 4 n | 4 % | 5 n | 5 % | Weighted average |
|---|---|---|---|---|---|---|---|---|---|---|---|---|
| Case 4 | 141 | 0 | 0.0 | 3 | 2.1 | 8 | 5.7 | 42 | 29.8 | 88 | 62.4 | 4.5 |
| Case 6 | 122 | 0 | 0.0 | 3 | 2.5 | 12 | 9.8 | 36 | 29.5 | 71 | 58.2 | 4.4 |
| Case 7 | 118 | 0 | 0.0 | 1 | 0.8 | 7 | 5.9 | 30 | 25.4 | 80 | 67.8 | 4.6 |
| Case 8 | 103 | 0 | 0.0 | 7 | 6.8 | 17 | 16.5 | 34 | 33.0 | 44 | 42.7 | 4.1 |
| Case 9 | 99 | 0 | 0.0 | 6 | 6.1 | 11 | 11.1 | 36 | 36.4 | 46 | 46.5 | 4.2 |
| Case 10 | 97 | 2 | 2.1 | 4 | 4.1 | 11 | 11.3 | 28 | 28.9 | 51 | 52.6 | 4.3 |

**Demand of cases**

| Case | Students answering n | Average % |
|---|---|---|
| Case 1 | 166 | 55 |
| Case 2 | 155 | 54 |
| Case 3 | 139 | 56 |
| Case 4 | 141 | 51 |
| Case 5 | 105 | 55 |
| Case 6 | 122 | 57 |
| Case 7 | 117 | 50 |
| Case 8 | 105 | 60 |
| Case 9 | 98 | 44 |
| Case 10 | 98 | 52 |

Numbers of students answering each question are shown and respective % of total for each individual case. Questions were based on a 5-point Likert scale (1 = strongly disagree to 5 = strongly agree).

working through the cases without watching the lectures was not possible. the cases are an incentive for watching the lectures").

The **practical context** of the cases helped to make learning more realistic (S13, C1: "The case places you directly into clinical routine. You have really the feeling to oversee the case. The proximity to reality is present is this concept". Embedding of clinical pictures and features made it realistic; S31, C1: "Through pictures everything was more realistic"). Some students noted that practical case-based learning helped them to remember the clinical context especially when practical courses could not be held (S9, C4: "practical case-based learning helps me remembering especially when practical courses are cancelled", S24, C24: "It is a good "alternative" for bedside teaching, as you can interactively learn with cases close to reality. That revives enjoying medicine").

**Enjoyment and motivation** aspects were also highlighted by students as learning aids. Presentation of cases like a game was entertaining and stimulated motivation (S85, C1 "Presentation like a game, high motivation and fun!", S40, C2 "Super motivating at the end of the week"). The cases worked as a motivational booster for students to work through the lectures at the end of the week (S55, C4 "This motivates to work through the lectures in time until the end of the week", S28, C2 "This format is great and both helpful, motivating and instructive").

## Cases as drivers for understanding of clinical work-flows and self-dependent learning in retrospect

Over all 51 (25.8%) students returned the online questionnaire after the completion of all 10 cases. Most students evaluated the cases all over as very helpful to understand clinical work-flows. Students were looking forward to work through the cases (Table 2) and performance of all cases as learning control in retrospect showed good results (Table 2).

**Table 2. Evaluation results after all cases in retrospect.**

| | Students answering | Likert Scale | | | | | | | | | | Weighted average |
|---|---|---|---|---|---|---|---|---|---|---|---|---|
| | n | 1 | | 2 | | 3 | | 4 | | 5 | | |
| | | n | % | n | % | n | % | n | % | n | % | |
| **Anticipation** | 50 | 0 | 0.0 | 2 | 4.0 | 11 | 22.0 | 20 | 40.0 | 17 | 34.0 | 4.4 |
| **Improvement of knowledge of clinical workflows** | 50 | 0 | 0.0 | 4 | 8.0 | 12 | 24.0 | 15 | 30.0 | 19 | 38.0 | 3.9 |
| **Performance of cases as learning control in retrospect** | | | | | | | | | | | | |
| **Case** | Students answering | Likert Scale | | | | | | | | | | Weighted average |
| | n | 1 | | 2 | | 3 | | 4 | | 5 | | |
| | | n | % | n | % | n | % | n | % | n | % | |
| Case 1 | 46 | 0 | 0.0 | 0 | 0.0 | 7 | 15.2 | 26 | 56.5 | 13 | 28.3 | 4.1 |
| Case 2 | 46 | 0 | 0.0 | 0 | 0.0 | 5 | 10.9 | 24 | 52.2 | 17 | 37.0 | 4.3 |
| Case 3 | 46 | 0 | 0.0 | 0 | 0.0 | 9 | 19.6 | 23 | 50.0 | 14 | 30.4 | 4.1 |
| Case 4 | 46 | 0 | 0.0 | 1 | 2.2 | 6 | 13.0 | 25 | 54.3 | 14 | 30.4 | 4.1 |
| Case 5 | 46 | 1 | 2.2 | 2 | 4.3 | 10 | 21.7 | 21 | 45.7 | 12 | 26.1 | 3.9 |
| Case 6 | 46 | 0 | 0.0 | 0 | 0.0 | 8 | 17.4 | 21 | 45.7 | 17 | 37.0 | 4.2 |
| Case 7 | 46 | 0 | 0.0 | 0 | 0.0 | 3 | 6.5 | 23 | 50.0 | 20 | 43.5 | 4.4 |
| Case 8 | 46 | 1 | 2.2 | 0 | 0.0 | 10 | 21.7 | 25 | 54.3 | 10 | 21.7 | 3.9 |
| Case 9 | 46 | 0 | 0.0 | 0 | 0.0 | 7 | 15.2 | 21 | 45.7 | 18 | 39.1 | 4.2 |
| Case 10 | 46 | 0 | 0.0 | 1 | 2.2 | 7 | 15.2 | 18 | 39.1 | 20 | 43.5 | 4.2 |

Students anticipation and knowledge of clinical workflows and cases as learning control. Questions were based on a 5-point Likert scale (1 = strongly disagree to 5 = strongly agree).

## Time spent with cases and beyond

**Fig 1. Approximate time spent with cases and for additional research triggered by the cases ($n$ = 46–51).** Data is presented as box plots with dots representing original data.

Students spent between 5 and 75 minutes (median 35) for each case (Fig 1). Students were triggered to search for additional information and contents on the topics and spent between 5 and 60 min (median 15) for additional research for each case respectively.

## Discussion

During the COVID-19 pandemic, students experienced social isolation, cancellation of practical training and a general loss of motivation. The challenges of teaching arising out of the unprecedented situation were decisive drivers behind the creation of new innovative e-learning cases. Efforts were made to maintain quality of medical education [3] and to keep students mentally in contact with patients in their endeavor to become physicians via e-learning cases. Early results on students' perception of e-learning reported optimism about the learning experience [24]. however, students did not prefer e-teaching over face-to-face teaching [9] and most of them preferred returning to the clinical setting [10]. Under these circumstances, factors that promoted identification with and working through presented materials on e-learning platforms gained invaluable importance. E-learning experiences should be designed with relevant and authentic information for the learners and should include intuitive navigation systems. Feedback mechanisms that are meaningful and adaptive, if possible, should be incorporated into the experience [25]. Previous reports about motivation techniques in e-learning by Tarans suggested 10 techniques for keeping students' attention: Stimuli, anticipation, incongruity, concreteness, variability, humor, inquiry, participation, breaks and energizers and storytelling. These were regarded as important elements to obtain and keep motivation on a high level while learning online [26].

In this study, we piloted an online medical teaching protocol using multimodal, game-based and realistic e-learning patient cases to replace in-person clinical teaching during the COVID-19 pandemic. Interest, motivation and learning goals of students were evaluated

through their responses received after every case during a 10-week period. At first sight, students were extrinsically motivated to participate in e-learning since it was mandatory to participate to be eligible for sitting the final exam. However, our findings suggest that students' motivation arose while solving the cases because of the reality-based settings, gaming-based simulations, appealing case designs and the in-case feedback mechanisms (Table 1). It has been reported previously that case design should be taken into consideration when promoting self-directed learning [27], which was reflected by our student cohort. E-interacting with patients and their relatives in these simulated scenarios helped students' motivation and proved reassuring for them during this mainly e-learning based semester. A real-world like e-learning experience showed students the practical relevance of the theories they were learning and made them easier to learn by forming associations between clinical presentations and learned theory.

Designing more realistic e-learning cases with multimedia elements as real-life scenarios helped to enhanced students' learning by making disease conditions more perspicuous. The cases provided a framework for students to understand clinical practice and routine, and allowed them to directly undergo clinicians' workflows by working through different patients' stories from a doctor's perspective. The cases also helped them develop clinical ways of thinking and to memorize the patients in the cases as"memorable patients"[28].

Moreover, these interactive aspects led to an enjoyable case experience (Table 1). Students often reported "fun" during the e-learning cases which corresponds with prior results were game-like e-learning proved to be more effective than conventional methods [29]. On the basis of these motivational aspects, the cases developed a much more important role during the course of the semester. Motivated by the setting, design and structure of the cases, students started to prepare better for the weekly case in order to fulfill their own ambition to serve the virtual patient sufficiently. This led to a pattern of weekly preparation, following and preparing the cases. This pattern is especially relevant in times of a pandemic as it helps the student to gain a structured learning week although no regular teaching was in place. Previous studies [30, 31] showed that motivation during self-directed learning while e-learning was the best predictor of changes in motivation.

This structuring aspect of the weekly cases supports the aims of competence-based internal medicine curriculum for building bridges between theoretical knowledge and practical skills and their application in an actual hospital setting. The students gradually started to perceive and understand the curriculum structure by the leading symptom-based cases, which helped them to better relate to the curriculum structure and philosophy.

## Limitation

Despite the interesting findings, the results should be interpreted in light of some limitations arising from the single-center study design. A cohort of 198 students at the Heidelberg University medical school enrolled in the internal medicine semester participated in the evaluations and only 25% of students participated in the final evaluation. Moreover, general effects of COVID-19 and social distancing must be noted: Highly motivated at the beginning, students' participation in voluntary evaluations declined throughout the 10-week curriculum and also the length of free-text answers reduced.

## Conclusions

This qualitative single-center study reports on integrating realistic symptom-based, e-learning cases in a completely online-based internal medicine curriculum at Heidelberg University Medical School by assessing motivational aspects regarding self-centered learning in a large

student cohort. Interactivity, media and design, repetition and deepening of knowledge, practical real-life aspects and fun were named by students as critical points for e-learning success.

## Supporting information

**S1 Text. Supplementary methods.**
(DOCX)

**S2 Text. Original SurveyMonkey questionnaires [German].**
(DOCX)

**S3 Text. Original SurveyMonkey questionnaires [English].**
(DOCX)

**S1 Table. Design of HeiCuMed internal medicine curriculum.**
(DOCX)

**S2 Table. Case overview of embedded interactive tools.**
(DOCX)

**S1 Video.**
(TXT)

**S2 Video.**
(TXT)

**S1 Share link. Case 1 –chest pain [Engl. version].**
(TXT)

**S2 Share link. Case 6—edema [Engl. version].**
(TXT)

**S1 Graphical abstract.**
(TIF)

## Acknowledgments

We acknowledge the support from Heike Ruck, Inken Macht and Larissa Schönhoff regarding communication with students and support with HeiCuMed e-learning platform Moodle. We thank Jörd Rodrian for support in preparation of the video files for case presentation in the supplements. Furthermore, we thank all medical students for their participation and feedback.

## Author Contributions

**Conceptualization:** Ann-Kathrin Rahm, Christoph Nikendei, Jobst-Hendrik Schultz.

**Data curation:** Ann-Kathrin Rahm, Maximilian Töllner, Max Ole Hubert, Katrin Klein, Cyrill Wehling, Tim Sauer, Hannah Mai Hennemann, Selina Hein, Zoltan Kender, Janine Günther, Petra Wagenlechner, Till Johannes Bugaj.

**Formal analysis:** Ann-Kathrin Rahm, Sophia Boldt.

**Funding acquisition:** Ann-Kathrin Rahm, Christoph Nikendei, Jobst-Hendrik Schultz.

**Investigation:** Ann-Kathrin Rahm, Maximilian Töllner, Max Ole Hubert, Jobst-Hendrik Schultz.

**Methodology:** Ann-Kathrin Rahm, Maximilian Töllner, Max Ole Hubert, Jobst-Hendrik Schultz.

**Project administration:** Ann-Kathrin Rahm, Jobst-Hendrik Schultz.

**Resources:** Ann-Kathrin Rahm, Katrin Klein, Cyrill Wehling, Tim Sauer, Hannah Mai Hennemann, Selina Hein, Zoltan Kender, Janine Günther, Petra Wagenlechner, Till Johannes Bugaj, Christoph Nikendei, Jobst-Hendrik Schultz.

**Software:** Ann-Kathrin Rahm.

**Supervision:** Ann-Kathrin Rahm, Christoph Nikendei, Jobst-Hendrik Schultz.

**Validation:** Ann-Kathrin Rahm.

**Visualization:** Ann-Kathrin Rahm, Maximilian Töllner, Max Ole Hubert.

**Writing – original draft:** Ann-Kathrin Rahm.

**Writing – review & editing:** Ann-Kathrin Rahm, Maximilian Töllner, Max Ole Hubert, Katrin Klein, Cyrill Wehling, Tim Sauer, Hannah Mai Hennemann, Selina Hein, Zoltan Kender, Janine Günther, Petra Wagenlechner, Till Johannes Bugaj, Sophia Boldt, Christoph Nikendei, Jobst-Hendrik Schultz.

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
