## [Decision Letter · Decision Letter 0]

11 Feb 2021

PONE-D-20-40841

Effects of Realistic e-Learning Cases on Students` Learning Motivation during COVID-19

PLOS ONE

Dear Dr. Rahm,

Thank you for submitting your manuscript to PLOS ONE. After careful consideration, we feel that it has merit but does not fully meet PLOS ONE’s publication criteria as it currently stands. Therefore, we invite you to submit a revised version of the manuscript that addresses the points raised during the review process.

We look forward to receiving your revised manuscript.

Kind regards,

Prof, Mojtaba Vaismoradi, PhD, MScN, BScN

Academic Editor

PLOS ONE

Journal Requirements:

2.Thank you for stating the following in the Competing Interests section:

"AKR carried out this project for her Master of Medical Education degree. "

Reviewers' comments:

Reviewer #1: Dear author,

I enjoyed reading your paper, however minor revision is warranted.

The introduction needs expanding in order to give more background on the topic. I felt it was too short.

The methodology is good however more elaboration and references shall be added in the design of the paper and sampling technique.

How did you calculate the sample size?

How did you maintain ethical standards ? elaborate.

I prefer adding tables to your paper which makes it more readable and reader friendly than the figures used.

Reviewer #2: # PONE-D-20-40841 entitled "Effects of Realistic e-Learning Cases on Students` Learning Motivation during COVID- 19" for the PLOS ONE.

The authors presented a study about real-life patient e-learning modules to transmit competency-based learning contents to medical students and evaluated their responses about their experience. However, the study still needs to be clarified:

1- Although the study is interesting, only 164 students answered the survey and only 25% of this amount made the final test of mandatory activity. What do the authors report the lack of motivation of students to complete a mandatory activity? Why did those who did like it? These points still need to be clearer.

2- Standardize the terms in the manuscript, for example "bed-side” and “COVID-19”.

Reviewer #3: This is an interesting study and the authors have collected a unique dataset using cutting edge methodology. The paper is generally well written and structured. The findings of a research are appealing and do really contribute to advancement significantly during and post COVID19 pandemic.

This paper has a potential to be accepted, but few points/typos have to be clarified or fixed before we can proceed further.

Page no. 8, line 12 - "Moreover, these interactive aspects led to an enjoyable case experience (Fig. 1 H)" - Is there a fig 1H. If so, please provide.

Supplementary Page no. 5. For example

Dyspnoe should be amended as dyspnoea

Szenario should be amended as scenario.

Pls make sure that whether to use British English or German but are consistent.

Reviewer #4: The research was timing and interesting, the claimed result seems acceptable and up to the mark, the work shows its originality but slightly weak in technical support. However, the authors were advised to consider the following few comments in the current version.

1. The result section should contain research output from the applied methodology, please review the structure of the Results and methods section for the same. Considerable information should be given.

2. Our readers will get benefitted from the structure (In English-to add in the paper) of the online survey performed by SurveyMonkey.

3. Figure need revision, improve its visibility and scale especially Figure 1.

4. It is strictly advised to add more technical support (like Statistical analysis) to the claimed results for such valuable work

5. The research was Interesting.

---

## [Author Response · Author response to Decision Letter 0]

26 Feb 2021

Response to Reviewers

PONE-D-20-40841

Effects of Realistic e-Learning Cases on Students` Learning Motivation during COVID-19

PLOS ONE

We would like to thank the reviewers for theri encouraging comments and valuable suggestions, leading us to revise and significantly improve our manuscript. Details are outlined below. Changes are highlighted in red in the revised manuscript. 

Reviewer #1: 

Dear author,

I enjoyed reading your paper, however minor revision is warranted.

We thank the reviewer for evaluating our paper and are happy that the experience of reading the manuscript was enjoyable. 

The introduction needs expanding in order to give more background on the topic. I felt it was too short.

Our wish was to supply a brief and concise introduction in these rattled times of the COVID-19 pandemic. We performed an additional current literature search and lengthened the introduction regarding aspects of e-learning during the current COVID-19 pandemic. 

The introduction was lengthened as follows:

“Online learning can show advantage by using student-centered approaches to facilitate educational access. Despite all the advantages, online learning is not the ultimate solution to all academic issues as in medical and other health professional courses, the main drawback of e-learning is in most occasions the impossibility to practice live11. 

For effectiveness of online learning principles of digital learning, goals, and student's preferences should be taken into consideration12. Various studies have assessed preparedness for purely e-Learning in different countries. In an analysis in Lybia only twelve percent of students agreed that e-learning could be used for clinical aspects and students were concerned about how e-learning could be applied to provide clinical experience13. Regarding social distancing and sole online learning effects on students` motivation are also important under these circumstances14 

Added references:

• Camargo CP, Tempski PZ, Busnardo FF, Martins MA, Gemperli R. Online learning and COVID-19: a meta-synthesis analysis. Clinics (Sao Paulo). 2020 Nov 6;75:e2286. doi: 10.6061/clinics/2020/e2286

• Jiang Z, Wu H, Cheng H, Wang W, Xie A, Fitzgerald SR. Twelve tips for teaching medical students online under COVID-19. Med Educ Online. 2021 Dec;26(1):1854066. doi: 10.1080/10872981.2020.1854066

• Alsoufi A, Alsuyihili A, Msherghi A, Elhadi A, Atiyah H, Ashini A, Ashwieb A, Ghula M, Ben Hasan H, Abudabuos S, Alameen H, Abokhdhir T, Anaiba M, Nagib T, Shuwayyah A, Benothman R, Arrefae G, Alkhwayildi A, Alhadi A, Zaid A, Elhadi M. Impact of the COVID-19 pandemic on medical education: Medical students' knowledge, attitudes, and practices regarding electronic learning. PLoS One. 2020 15(11):e0242905 doi: 10.1371/journal.pone.0242905

• Saiyad S, Virk A, Mahajan R, Singh T. Online Teaching in Medical Training: Establishing Good Online Teaching Practices from Cumulative Experience. Int J Appl Basic Med Res. 2020 Jul-Sep;10(3):149-155. doi: 10.4103/ijabmr.IJABMR_358_20

• Deepika V, Soundariya K, Karthikeyan K, Kalaiselvan G. 'Learning from home': role of e-learning methodologies and tools during novel coronavirus pandemic outbreak. Postgrad Med J. 2020 postgradmedj-2020-137989. doi: 10.1136/postgradmedj-2020-137989

The methodology is good however more elaboration and references shall be added in the design of the paper and sampling technique.

How did you calculate the sample size?

We thank the reviewer for approving our methodology. As wished, references for methodology were added and the section on the qualitative data analysis of free-text answers was extended. Sample size for this qualitative and descriptive analysis arose from the total number of students enrolled in the internal medicine course during summer semester 2020. As performing the cases was mandatory all students were invited to take part in the facultative evaluations. 

Added references for methodology: 

• Mayring P. Qualitative Inhaltsanalyse. Grundlagen und Techniken (7. Auflage, erste Auflage 1983) Deutscher Studien Verlag: Weinheim; 2000

• Strauss A, Corbin J. Basics of qualitative research: techniques and procedures for developing grounded theory. Wiesbaden: Sage publications, Inc; 1998

The methods section on data analysis was supplemented as follows: 

„Data analysis was performed with OriginPro2020 and MaxQDA2020 (VERBI GmbH). As free-text answers were already in written form, no transcription was needed. Texts obtained from SurveyMonkey results were checked for information that could identify an individual student. As no individuals could be identified, the interviews were subjected to a qualitative content analysis following Mayring‘s principles of inductive category development22. First, we undertook an open coding of all free-text answer questions to identify possible recurring topics. Next, individual sentences or passages were identified as one code, representing the most elementary unit of the resulting protocol23. Exemplary students’ quotes were translated into English and embedded within the text. Using the software MaxQDA (version 2020, VERBI Software - Consult - Social Research GmbH, Berlin), we summarized individual codes as relevant topics for each case. Then, analyzers compared recurring topics from the individual cases results and assigned them to higher-level categories. The respective codes and topics were discussed to reach consensus (investigator triangulation). Finally, we subsumed the topics into a total of n = 6 relevant categories. We applied the categories to all transcripts using the software MaxQDA.”

How did you maintain ethical standards? elaborate.

To obtain long-free text answers by our student collective, a true anonymization was performed via Survey Monkey and not via the regular evaluation platform EvaSys which allows personal tracking of students. Hence ethical standards were maintained firstly by performing true anonymous evaluation via survey monkey. Secondly, for the retrospective analysis of data, an ethics approval from Heidelberg University ethics committee was obtained before performing analysis. 

I prefer adding tables to your paper which makes it more readable and reader friendly than the figures used.

We appreciate the reviewer’s suggestion for improving readability of our manuscript. As requested the figures were replaced by tables containing the same original data sets and information. Figure 1 and 2 were mainly replaced by tables. Only the original Figure 2C was kept as Figure 1 as visualization of spent time appeared clearer in the graph than a table and box plots with underlying dots of original data points also make the underlying distribution obvious. 

Figure 1

Approximate time spent with cases and for additional research triggered by the cases (n=46-51). Data is presented as box plots with dots representing original data. 

Table 1

Case Design

Case Students 

answering

n Likert Scale Weighted 

average

 1 2 3 4 5 

 n % n % n % n % n % 

Case 1 164 0 0.0 0 0.0 0 0.0 21 12.8 144 87.8 4.9

Case 2 154 0 0.0 0 0.0 2 1.3 51 33.1 101 65.6 4.7

Case 3 139 3 2.2 7 5.0 41 29.5 45 32.4 43 30.9 3.9

Case 4 141 0 0.0 1 0.7 12 8.5 40 28.4 88 62.4 4.5

Case 5 106 0 0.0 3 2.8 21 19.8 44 41.5 38 35.8 4.1

Case 6 121 0 0.0 1 0.8 10 8.3 39 32.2 71 58.7 4.5

Case 7 117 0 0.0 2 1.7 6 5.1 31 26.5 78 66.7 4.6

Case 8 103 1 1.0 4 3.9 17 16.5 33 32.0 48 46.6 4.2

Case 9 99 0 0.0 5 5.1 15 15.2 39 39.4 40 40.4 4.2

Case 10 98 0 0.0 2 2.0 12 12.2 31 31.6 53 54.1 4.4

Navigation

Case Students 

answering

n Likert Scale Weighted 

average

 1 2 3 4 5 

 n % n % n % n % n % 

Case 1 166 0 0.0 1 0.6 5 3.0 25 15.1 135 81.3 4.8

Case 2 153 0 0.0 0 0.0 2 1.3 20 13.1 131 85.6 4.8

Case 3 139 0 0.0 0 0.0 3 2.2 16 11.5 120 86.3 4.8

Performance of quiz modi

Case Students 

answering

n Likert Scale Weighted 

average

 1 2 3 4 5 

 n % n % n % n % n % 

Case 1 166 1 0.6 7 4.2 29 17.5 37 22.3 92 55.4 4.3

Case 2 155 0 0.0 2 1.3 7 4.5 22 14.2 124 80.0 4.7

Case 3 139 0 0.0 1 0.7 3 2.2 13 9.4 122 87.8 4.8

Case 4 141 0 0.0 9 6.4 22 15.6 44 31.2 66 46.8 4.2

Case 5 106 0 0.0 2 1.9 12 11.3 18 17.0 74 69.8 4.6

Case 6 122 3 2.5 3 2.5 14 11.5 29 23.8 73 59.8 4.4

Case 7 118 2 1.7 1 0.8 3 2.5 13 11.0 99 83.9 4.8

Case 8 103 2 1.9 2 1.9 14 13.6 34 33.0 51 49.5 4.3

Case 9 99 0 0.0 1 1.0 12 12.1 25 25.3 61 61.6 4.5

Case 10 98 0 0.0 4 4.1 4 4.1 20 20.4 70 71.4 4.6

Performance as learning control 

Case Students 

answering

n Likert Scale Weighted 

average

 1 2 3 4 5 

 n % n % n % n % n % 

Case 1 166 1 0.6 3 1.8 8 4.8 41 24.7 113 68.1 4.6

Case 2 155 0 0.0 1 0.6 5 3.2 28 18.1 121 78.1 4.7

Case 3 139 8 5.8 14 10.1 46 33.1 33 23.7 38 27.3 3.6

Case 4 141 0 0.0 3 2.1 11 7.8 45 31.9 82 58.2 4.5

Case 5 106 1 0.9 1 0.9 24 22.6 25 23.6 55 51.9 4.3

Case 6 122 0 0.0 4 3.3 7 5.7 29 23.8 82 67.2 4.6

Case 7 118 0 0.0 0 0.0 10 8.5 22 18.6 86 72.9 4.6

Case 8 103 2 1.9 10 9.7 18 17.5 33 32.0 40 38.8 4.0

Case 9 98 3 3.1 10 10.2 20 20.4 29 29.6 36 36.7 3.9

Helpfulness of senior physician's comments

Case Students 

answering

n Likert Scale Weighted 

average

 1 2 3 4 5 

 n % n % n % n % n % 

Case 1 162 1 0.6 1 0.6 10 6.2 43 26.5 107 66.0 4.6

Case 2 154 0 0.0 1 0.6 2 1.3 27 17.5 124 80.5 4.8

Case 3 139 1 0.7 3 2.2 14 10.1 43 30.9 78 56.1 4.4

Case 4 141 0 0.0 6 4.3 12 8.5 38 27.0 85 60.3 4.4

Case 5 106 0 0.0 1 0.9 7 6.6 16 15.1 82 77.4 4.7

Case 6 122 0 0.0 2 1.6 6 4.9 24 19.7 90 73.8 4.7

Case 7 118 0 0.0 2 1.7 5 4.2 25 21.2 86 72.9 4.7

Case 8 102 2 2.0 1 1.0 7 6.9 33 32.4 59 57.8 4.4

Case 9 99 1 1.0 0 0.0 7 7.1 25 25.3 66 66.7 4.6

Case 10 98 0 0.0 1 1.0 8 8.2 22 22.4 67 68.4 4.6

Fun 

Case Students 

answering

n Likert Scale Weighted 

average

 1 2 3 4 5 

 n % n % n % n % n % 

Case 4 141 0 0.0 3 2.1 8 5.7 42 29.8 88 62.4 4.5

Case 6 122 0 0.0 3 2.5 12 9.8 36 29.5 71 58.2 4.4

Case 7 118 0 0.0 1 0.8 7 5.9 30 25.4 80 67.8 4.6

Case 8 103 0 0.0 7 6.8 17 16.5 34 33.0 44 42.7 4.1

Case 9 99 0 0.0 6 6.1 11 11.1 36 36.4 46 46.5 4.2

Case 10 97 2 2.1 4 4.1 11 11.3 28 28.9 51 52.6 4.3

Demand of cases

Case Students 

answering

n Average 

 %

Case 1 166 55

Case 2 155 54

Case 3 139 56

Case 4 141 51

Case 5 105 55

Case 6 122 57

Case 7 117 50

Case 8 105 60

Case 9 98 44

Case 10 98 52

Table 1

Case-to-case evaluation by students for structure, navigation, quiz functions, performance as learning control, helpfulness of senior physician’s comments, aspect of fun and demand. Numbers of students answering each question are shown and respective % of total for each individual case. Questions were based on a 5-point Likert scale (1=strongly disagree to 5=strongly agree).

Table 2

 Students 

answering

n Likert Scale Weighted 

average

 1 2 3 4 5 

 n % n % n % n % n % 

Anticipation 50 0 0.0 2 4.0 11 22.0 20 40.0 17 34.0 4.4

Improvement of knowledge

of clinical workflows 50 0 0.0 4 8.0 12 24.0 15 30.0 19 38.0 3.9

Performance of cases as learning control in retrospect 

Case Students 

answering

n Likert Scale Weighted 

average

 1 2 3 4 5 

 n % n % n % n % n % 

Case 1 46 0 0.0 0 0.0 7 15.2 26 56.5 13 28.3 4.1

Case 2 46 0 0.0 0 0.0 5 10.9 24 52.2 17 37.0 4.3

Case 3 46 0 0.0 0 0.0 9 19.6 23 50.0 14 30.4 4.1

Case 4 46 0 0.0 1 2.2 6 13.0 25 54.3 14 30.4 4.1

Case 5 46 1 2.2 2 4.3 10 21.7 21 45.7 12 26.1 3.9

Case 6 46 0 0.0 0 0.0 8 17.4 21 45.7 17 37.0 4.2

Case 7 46 0 0.0 0 0.0 3 6.5 23 50.0 20 43.5 4.4

Case 8 46 1 2.2 0 0.0 10 21.7 25 54.3 10 21.7 3.9

Case 9 46 0 0.0 0 0.0 7 15.2 21 45.7 18 39.1 4.2

Case 10 46 0 0.0 1 2.2 7 15.2 18 39.1 20 43.5 4.2

Table 2

Evaluation results after all cases in retrospect. Students anticipation and knowledge of clinical workflows and cases as learning control (n = 46-50). Questions were based on a 5-point Likert scale (1=strongly disagree to 5=strongly agree).

Reviewer #2: # PONE-D-20-40841 entitled "Effects of Realistic e-Learning Cases on Students` Learning Motivation during COVID- 19" for the PLOS ONE.

The authors presented a study about real-life patient e-learning modules to transmit competency-based learning contents to medical students and evaluated their responses about their experience. However, the study still needs to be clarified:

1- Although the study is interesting, only 164 students answered the survey and only 25% of this amount made the final test of mandatory activity. What do the authors report the lack of motivation of students to complete a mandatory activity? Why did those who did like it? These points still need to be clearer.

We thank the reviewer for valuing our study as interesting and suggesting clarification of our manuscript regarding mandatory and facultative aspects. 

Only performing the cases itself was mandatory for the students. Participating in the evaluations, both the individual case evaluations and the final evaluation, was facultative from the beginning till the end of the semester. The evaluations of individual cases were not mandatory but directly linked at the end of the mandatory cases which may explain higher response rates in these individual case evaluations compared to the final evaluation. This final survey was performed one week after the mandatory internal medicine MCQ test which may potentially explain students’ reduced motivation to participate. Also participation in the final evaluation could be less because it was not directly linked to a mandatory exercise. 

The methods section was clarified as follows: 

“As performing the cases was mandatory, all students were invited to take part in the facultative evaluations directly after the cases. An evaluation of all cases in retrospect was performed at the end of the semester voluntarily.”

2- Standardize the terms in the manuscript, for example "bed-side” and “COVID-19”.

Thank you for this suggestion. We checked the terms in the manuscript and standardized their spelling. 

Reviewer #3: 

This is an interesting study and the authors have collected a unique dataset using cutting edge methodology. The paper is generally well written and structured. The findings of a research are appealing and do really contribute to advancement significantly during and post COVID19 pandemic.

This paper has a potential to be accepted, but few points/typos have to be clarified or fixed before we can proceed further.

Page no. 8, line 12 - "Moreover, these interactive aspects led to an enjoyable case experience (Fig. 1 H)" - Is there a fig 1H. If so, please provide.

We thank the reviewer for detecting this typing error. As figures were mostly removed by tables as requested by reviewer#1 this reference was set to “(Table 1)”. 

Supplementary Page no. 5. For example

Dyspnoe should be amended as dyspnoea

Szenario should be amended as scenario.

Pls make sure that whether to use British English or German but are consistent.

We thank the reviewer for highlighting this spelling errors. The changes in the new S2 Table have been performed accordingly. Usage of American English was checked throughout the manuscript. Only the supplemental document containing SurveyMonkey questions was left in German, as students received those original questions in German. A translated version of these questions can be now found also in the supplements (S3 text)

The captions of the supplemental documents have been included in the main manuscript. 

„S2 Text - Original Survey Monkey Questionnaires [in German]

S3 Text – Original Survey Monkey Questionnaires [translated to English]”

Reviewer #4: 

The research was timing and interesting, the claimed result seems acceptable and up to the mark, the work shows its originality but slightly weak in technical support. However, the authors were advised to consider the following few comments in the current version.

1. The result section should contain research output from the applied methodology, please review the structure of the Results and methods section for the same. Considerable information should be given.

As requested by the reviewer the results and methods section were reviewed for strict separation of contents and n numbers were removed from the methods section and solely presented in the results section. We hope the changes find the reviewer’s approval. 

2. Our readers will get benefitted from the structure (In English-to add in the paper) of the online survey performed by SurveyMonkey.

We appreciate the reviewer’s suggestion for clarification for a purely English-speaking readership. The Original German version was left in to demonstrate also the original German questions that were asked. A translated version of these SurveyMonkey questions can be now found also in the supplements (S3 text).

The captions of the supplemental documents have been included in the main manuscript. 

„S3 Text – Original Survey Monkey Questionnaires [translated to English]”

3. Figure need revision, improve its visibility and scale especially Figure 1.

As requested by reviewer #1 most Figures in the manuscript have been replaced by tables containing the same data sets for improving readability of the manuscript. We hope these changes will also obtain approval from reviewer #4. 

4. It is strictly advised to add more technical support (like Statistical analysis) to the claimed results for such valuable work

The newly supplied tables that replaced mainly Figure 1 and 2 resulting data from our SurveyMonkey questions are now presented in more detail. As questions were either asked in Likert Scale questions or in free-text answers only limited quantitative analysis can be performed. Likert Scale answers were supplemented with percentage distributions to add more statistical analysis (Table 1 and 2). 

For the free-text answers a distributional analysis and visualization can be performed by word clouds. As our student’s answers for the free-text answer questions were in German we did not present this word cloud in the manuscript and a direct translation would not have been methodologically correct. 

For review only – 

Word cloud presentation regarding frequencies of used German words in student’s free-text answers. 

5. The research was Interesting.

We thank the reviewer for his/her evaluation of our work and the requested changes that helped us to even make the manuscript more concise and interesting.

---

## [Decision Letter · Decision Letter 1]

18 Mar 2021

Effects of Realistic e-Learning Cases on Students` Learning Motivation during COVID-19

PONE-D-20-40841R1

Dear Dr. Rahm,

We’re pleased to inform you that your manuscript has been judged scientifically suitable for publication and will be formally accepted for publication once it meets all outstanding technical requirements.

Kind regards,

Prof, Mojtaba Vaismoradi, PhD, MScN, BScN

Academic Editor

PLOS ONE

---

## [Editor Report · Acceptance letter]

5 Apr 2021

PONE-D-20-40841R1 

Effects of Realistic e-Learning Cases on Students` Learning Motivation during COVID-19 

Dear Dr. Rahm:

I'm pleased to inform you that your manuscript has been deemed suitable for publication in PLOS ONE. Congratulations! Your manuscript is now with our production department. 

Kind regards, 

on behalf of

Professor Mojtaba Vaismoradi 

Academic Editor

PLOS ONE